# Physical activity, balance, and bicycling in older adults

Maya Baughn[1], Victor Arellano[1]*, Brieanna Hawthorne-Crosby[2‡], Joseph S. Lightner[1‡], Amanda Grimes[1‡], Gregory King[2]

1 School of Nursing and Health Studies, University of Missouri-Kansas City, Kansas City, Missouri, United States of America, 2 Division of Energy, Matter and Systems, School of Science and Engineering, University of Missouri-Kansas City, Kansas City, Missouri, United States of America

☯ These authors contributed equally to this work.
‡ BHC, JSL and AG also contributed equally to this work.
* vracpc@umsystem.edu

**Data Availability Statement:** All relevant data are within the paper and its Supporting information files.

**Funding:** The project was funded by the University of Missouri-Kansas City's Office of Undergraduate

## Abstract

Falls are a critical public health issue among older adults. One notable factor contributing to falls in older adults is a deterioration of the structures supporting balance and overall balance control. Preliminary evidence suggests older adults who ride a bicycle have better balance than those who do not. Cycling may be an effective intervention to prevent falls among older adults. This study aims to objectively measure the relationship between bicycling, physical activity, and balance for older adults. Older adult cyclists (n = 19) and non-cyclists (n = 27) were recruited to (1) complete a survey that assessed demographics; (2) wear an accelerometer for 3 weeks to objectively assess physical activity; and (3) complete balance-related tasks on force platforms. Mann-Whitney U-tests were performed to detect differences in balance and physical activity metrics between cyclists and non-cyclists. Cyclists were significantly more physically active than non-cyclists. Cyclists, compared to non-cyclists, exhibited differences in balance-related temporospatial metrics and long-range temporal correlations that suggest a more tightly regulated postural control strategy that may relate to higher stability. Cycling was observed to correlate more strongly with balance outcomes than other physical activity. Taken together, these results demonstrate the possible implications for cycling as an effective intervention to improve balance and reduce fall risk.

## Introduction

Falls are a significant public health problem for older adults [1]. In the United States, $50 billion is spent annually on fall-related injuries in older adults [2]. With more than 35.6 million cases reported among older adults in 2018 [3], falls are the leading cause of injury for this age group. Additionally, the number of age-related falls and their related costs are expected to rise with the growth of the aging population [2]. An individual's risk of falling rises exponentially during older adulthood; reducing an individual's risk of falling is crucial for the growing population [4].

Research and Retirees Association. The funders had no role in study design, data collection and analysis, decision to publish, or preparation of the manuscript.

**Competing interests:** The authors have declared that no competing interests exist.

Reduced balance and lack of exercise are two closely related factors that lead to further health issues in older adults and significantly increase the risk for falls [5]. Additionally, obese older adults have a greater risk for falls due to reduced balance and range of motion [6]. Previous research has shown that aging can cause a decline in the physiologic structures supporting the body's ability to balance [7, 8], which can be a barrier to completing activities of daily living and physical activity [9]. Evidence suggests that between 20% and 50% of older adults have a balance impairment [7–9]. Older adults often underestimate their risk for falls, feeling that they are not balance impaired when they actually may have a moderate-to-high risk [10]. Therefore, objectively measuring balance may provide insights to fall prevention.

Risk for falls can be mitigated to some degree with exercise [4]. For individuals aged 65 years and older, the Physical Activity Guidelines for Americans recommends 150 minutes of moderate to intense activity per week, two days of muscle-strengthening, and the addition of balance training [11]. Many older adults do not meet this recommendation [12], particularly following decreases in physical activity levels associated with the COVID-19 pandemic [13]. Maintaining physical activity, particularly with exercises targeting balance like yoga or tai chi, is crucial for older adults to maintain healthy, independent lives [11].

Emerging evidence suggests that bicycling offers a unique way to develop or maintain balance for older adults. Cycling has gained popularity among older adults, leading to increased leg strength, balance, and improved cardiovascular circulation [14, 15]. A preliminary study [16] reported on a potential relationship between balance and bicycling activity for older adults. However, more rigorous research needs to be conducted to support potential cycling-based interventions to prevent falls in older adults.

While preliminary studies support exploring cycling and its relationship to reducing fall risk [15, 17], past studies have yet to examine balance differences among recreational cyclists and non-cyclists who are 65 years and older. Therefore, this study aims to objectively measure balance and physical activity among older cyclists and non-cyclists. We hypothesize that older adults who cycle will have better balance and engage in more physical activity than older adults who do not cycle.

## Materials and methods

All study procedures were approved by the University of Missouri-Kansas City Institutional Review Board (#2043565) with a Waiver of Documentation. Written consent was voluntarily obtained by all participants.

### Study design

**Participants.** Participants 65 years and older were recruited through social media, flyers at community-based organizations that serve older adults, and cycling groups in the Kansas City metropolitan area. Participants who bicycled at least once per week during non-winter months were assigned to the cycling group (N = 19). Participants who reported bicycling less than once per week during non-winter months, or not cycling at all, were assigned to the non-cycling group (N = 27). Each participant completed a single visit to the University of Missouri-Kansas City's Human Motion Laboratory to complete a survey and a balance assessment as well as receive a Garmin accelerometer watch. Upon completing all study procedures, participants were given a $25 gift card.

**Measures.** *Demographics*. Demographic information was collected electronically via a self-report survey. Variables included age, gender, race and/or ethnicity, income, level of education, and employment status.

*Physical activity*. Garmin Vivofit 4 accelerometers were used to measure objective physical activity. Participants regularly synced their watches to the Garmin application installed on their smartphones.

Self-reported data about frequency of various types of physical activity was collected using the Recent Physical Activity Questionnaire [18].

*Balance*. During their lab visit, participants completed the Tinetti Balance Assessment [19], a validated instrument designed to assess participants' fall risk based on a series of tasks, including sitting, standing, turning around, and walking. With the exception of the walking task, participants completed tasks with each foot in contact with a six-axis force platform (AMTI, Watertown, MA, USA). In addition to the Tinetti Balance Assessment tasks, each participant was asked to perform a single-leg stance trial on each leg for as long as possible up to 60 seconds.

During the Tinetti and single-leg stance tasks, participants' foot-floor ground reaction forces and moments were captured with the force platforms using a sampling frequency of 1000 Hz. While force platform data was captured during all tasks, only four tasks were used for further analysis. These included eyes-open and eyes-closed double-leg stance trials from the Tinetti assessment, and right and left single-leg stance trials. Double-leg stance trials were captured for a period of 30 seconds with one foot placed on each of two adjacent force platforms. Right and left single-leg stance trials were performed on the right and left force platforms, respectively. Single-leg stance trials were conducted for a maximum of 60 seconds and were ended if the participant's opposite foot touched the floor or if the participant required assistance from the research associate.

*Analysis plan*. Initial exploration of the demographic and outcome variables revealed non-normally distributed data for cycling and non-cycling groups. Therefore, independent samples Mann-Whitney U-tests were performed to investigate differences in continuous variables (i.e., age, physical activity, and balance) between cycling and non-cycling groups. Chi-square tests were used to assess differences in categorical variables between groups. A secondary analysis, a Pearson correlation, was performed to assess the relationships between types of physical activities and balance. Sport related physical activities from the Recent Physical Activity Questionnaire [18] were included in the correlation analysis, along with balance measures that differed significantly between cyclists and non-cyclists.

For their physical activity data to be included in the analysis, participants were required to wear accelerometers for a minimum of 8 hours between 9 am and 9 pm, and achieve at least 500 steps per day, for a total period of 3 weeks. Daily data were aggregated at the week level for each participant; a minimum of 1 day per week was required to be included in the analysis. The weekly data were then weighted ([mean of weekdays × 5] + [mean of weekend days × 2]) ÷ 7. When a participant did not wear the device for ≥1 weekday and ≥1 weekend day, the weekly value was calculated as a mean of all valid wear days for the week. This approach has been used in other studies examining physical activity behaviors [20, 21]. Moderate to vigorous physical activity was measured using the Garmin device's automated detection of activity. The device automatically measures active minutes when a user runs for at least 1 minute or walks for at least 10 consecutive minutes. Participants' physical activity data from the three weeks immediately following their visit to the lab was included in the analyses.

All force platform data was processed with MATLAB V20 (Mathworks, Natick, MA, USA). Various temporospatial measures of postural sway were extracted from center of pressure time histories, including sway area, average velocity, stability parameter levels, medial-lateral and anterior-posterior sway range and standard deviation, and task duration. Since double-limb stance trials were all 30 seconds in duration, the task duration outcome variable was only analyzed for single-leg stance trials. In addition to temporospatial measures, the time scaling

properties of medial-lateral and anterior-posterior center of pressure velocity were assessed using detrended fluctuation analysis (DFA), an approach first used by Peng et al. [22]. Briefly, DFA involves calculating the root-mean-square error (RMSE) between data points and least-squares lines fit to the corresponding data within increasingly larger time windows. If long-range temporal correlations are present, the resulting fluctuation function $R$ behaves according to a power law:

$$R(\tau) \sim \tau^{\alpha}$$

Where $\tau$ is the window size and $\alpha$ is a scaling exponent. The latter is defined as the slope of a least-squares line fit to $\log R(\tau)$ versus $\log \tau$. Scaling exponents close to 0.5 indicate the absence of temporal correlations (i.e. white noise), while values between 0 and 0.5 indicate anti-persistence, and values $>0.5$ indicate persistence [23–25]. When applied to COP velocity, this approach often reveals two distinct regions: a short-term region characterized by smaller window sizes and larger scaling exponents, followed by a long-term region with larger window sizes and smaller scaling exponents [25]. Accordingly, we calculated both short-term and long-term scaling exponents ($\alpha_{ST}$ and $\alpha_{LT}$) by fitting least-squares lines to the first and last third of $R(\tau)$, respectively. Time scaling properties of COP velocity were further characterized by determining the window size at which each bi-logarithmic fluctuation plot transitions from the short-term to the long-term region; this was defined as the window size $\tau_{Cross}$ at the inter-section point of the least-squares lines fit to short- and long-term regions. A representative plot of $\log \tau$ versus $\log R(\tau)$ is shown in Fig 1.

## Results

### Demographics and physical activity

Table 1 reports sample demographics and compares characteristics between cyclists and non-cyclists. Mean age for the study sample was 73.2 years with cyclists being significantly younger (69.11 years) than non-cyclists (76.11 years) (U = 111.00, p = .001). The sample was majority white (91.3%) and reported to be retired (80.4%). The most commonly reported income was less than $50,000 annually (50.0%). The most commonly reported education level was some college (32.6%). The total sample had an even distribution of males (47.5%) and females (50.0%). However, the cyclist group was majority male (84.2%) while the non-cyclist group was majority female (77.8%) ($X^2(4, N = 46) = 3.26, p < .001$). Additionally, cyclists achieved significantly more steps per day (6266.66) and moderate-to-vigorous physical activity (MVPA) per week (82.98) compared to non-cyclists (4353.51 and 47.61 respectively) (U = 102.00, p = .015 and U = 81.00, p = .002 respectively).

### Balance

Results comparing cyclists and non-cyclists balance are summarized in Table 2. No effect size is reported due to utilizing Mann-Whitney U tests. For eyes-closed stance trials, Mann-Whitney tests indicated that medial-lateral long-term scaling exponents among cyclists (mean = 0.306) were significantly smaller than those of non-cyclists (mean = 0.410) (U = 124.0, $n_{cyclist}$ = 19, $n_{non-cyclist}$ = 26, p = .005). For eyes-open stance trials, Mann-Whitney tests indicated that sway velocity among cyclists (mean = 12.05 mm/s) was significantly smaller than that among non-cyclists (mean = 13.86 mm/s) (U = 360.0 $n_{cyclist}$ = 19, $n_{non-cyclist}$ = 26, p = .009); and stability parameter levels among cyclists (mean = 15.67) were significantly smaller than those of non-cyclists (mean = 17.63) (U = 342.0, $n_{cyclist}$ = 19 $n_{non-cyclist}$ = 26, p = .029).

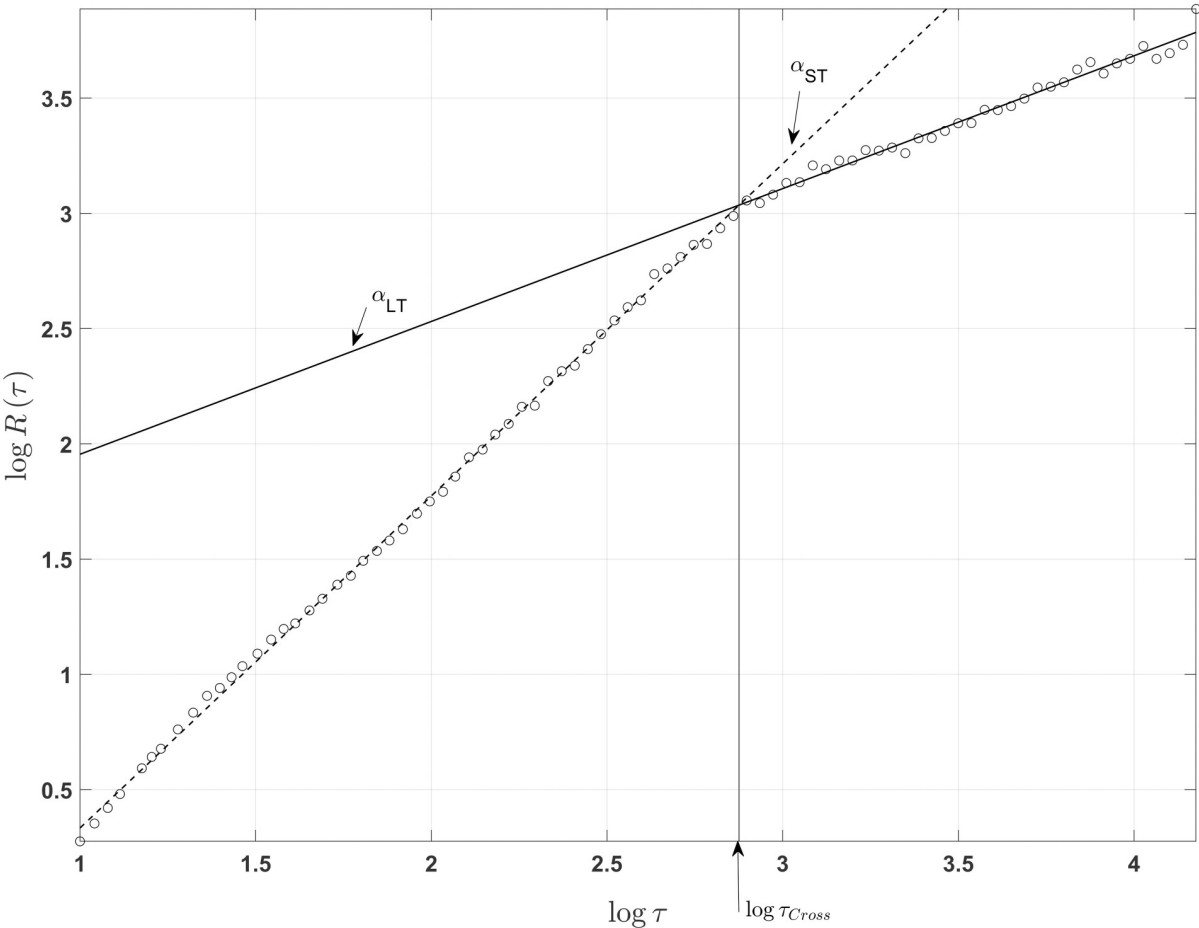

**Fig 1. Representative bi-logarithmic plot of DFA window size ($\tau$)versus fluctuation function $R(\tau)$.** For COP velocity, ($\tau$) tends to fall into short term and long term scaling regions characterized by slopes $\alpha_{ST}$ and $\alpha_{LT}$, respectively, and separated by crossover point $\tau_{Cross}$.

For right single-leg stance trials, Mann-Whitney tests indicated that the medial-lateral sway range among cyclists (mean = 49.37 mm) was significantly larger than that among non-cyclists (mean = 40.21 mm) (U = 301.0, $n_{cyclist}$ = 18 $n_{non\text{-}cyclist}$ = 23, p = .014); anterior-posterior sway range among cyclists (mean = 83.98 mm) was significantly larger than that among non-cyclists (mean = 67.12 mm) (U = 290.0, $n_{cyclist}$ = 18, $n_{non\text{-}cyclist}$ = 23, p = .029); task duration among cyclists (mean = 23.68 s) was significantly longer than that among non-cyclists (mean = 7.58 s) (U = 317.0, $n_{cyclist}$ = 18 $n_{non\text{-}cyclist}$ = 23, p = .004); medial-lateral long-term scaling exponents among cyclists (mean = 0.382) were significantly smaller than those among non-cyclists (mean = 0.724)(U = 82.0, $n_{cyclist}$ = 18, $n_{non\text{-}cyclist}$ = 23, p = .001); anterior-posterior long-term scaling exponents among cyclists (mean = 0.404) were significantly smaller than those among non-cyclists (mean = 0.775)(U = 103.0, $n_{cyclist}$ = 18, $n_{non\text{-}cyclist}$ = 23, p = .006); medial-lateral crossover points among cyclists (mean = 2.369) were significantly larger than those among non-cyclists (mean = 2.260)(U = 287.0, $n_{cyclist}$ = 18, $n_{non\text{-}cyclist}$ = 23, p = .036); and anterior-posterior crossover points among cyclists (mean = 2.374) were significantly larger than those among non-cyclists (mean = 2.275)(U = 307.0, $n_{cyclist}$ = 18, $n_{non\text{-}cyclist}$ = 23, p = .009).

For left single-leg stance trials, medial-lateral sway range among cyclists (mean = 43.55 mm) was significantly larger than that among non-cyclists (mean = 28.27 mm) (U = 93.00,

**Table 1. Comparison of demographics and characteristics.**

| | Total | | Cyclist | | Non-Cyclist | | | |
|---|---|---|---|---|---|---|---|---|
| | M or N | SD or % | M or N | SD or % | M or N | SD or % | U Statistic or $X^2$ | p-value |
| **Age** | 73.2 | 7.08 | 69.11 | 4.56 | 76.11 | 7.17 | 111.00 | < .001 |
| **Race** | | | | | | | 3.26 | .515 |
| White | 42 | 91.30 | 18 | 94.7 | 24 | 88.9 | | |
| Black | 2 | 4.30 | 1 | 5.3 | 1 | 3.7 | | |
| Multiple Races | 2 | 4.30 | 0 | 0 | 2 | 7.4 | | |
| **Gender** | | | | | | | 22.44 | .000 |
| Male | 21 | 45.7 | 16 | 84.2 | 5 | 18.5 | | |
| Female | 23 | 50 | 2 | 10.5 | 21 | 77.8 | | |
| Other | 2 | 4.3 | 1 | 5.3 | 1 | 3.7 | | |
| **Education** | | | | | | | 9.62 | .292 |
| < High School | 2 | 4.30 | 0 | 0 | 2 | 7.4 | | |
| High school graduate or GED | 7 | 15.20 | 2 | 10.5 | 5 | 18.5 | | |
| Some college | 15 | 32.60 | 3 | 15.80 | 12 | 44.4 | | |
| College graduate | 11 | 23.90 | 7 | 36.80 | 4 | 14.8 | | |
| Master's degree or higher | 11 | 23.90 | 7 | 36.80 | 4 | 14.8 | | |
| **Income** | | | | | | | 6.80 | .147 |
| <$50,000 | 23 | 50.00 | 5 | 26.30 | 18 | 66.70 | | |
| $50,000-<$100,000 | 14 | 30.40 | 7 | 36.80 | 7 | 25.90 | | |
| >$100,000 | 9 | 19.60 | 7 | 36.80 | 2 | 7.40 | | |
| **Employment** | | | | | | | 4.12 | .128 |
| Retired | 37 | 80.4 | 13 | 68.4 | 24 | 88.9 | | |
| Working | 9 | 19.6 | 6 | 31.6 | 3 | 11.1 | | |
| **Physical Activity** | | | | | | | | |
| Steps | 5187.45 | 2502.07 | 6266.66 | 2445.34 | 4353.51 | 2258.51 | 102.00 | .015 |
| Active Minutes | 63.03 | 36.56 | 82.98 | 41.41 | 47.61 | 23.25 | 81.00 | .002 |

Note: M = mean, N = sample size, SD = standard deviation

$n_{cyclist}$ = 18 $n_{non-cyclist}$ = 23, p = .003); anterior-posterior sway range among cyclists (mean = 78.02 mm) was significantly larger than that among non-cyclists (mean = 58.63 mm) (U = 121.0, $n_{cyclist}$ = 18 $n_{non-cyclist}$ = 23, p = .024); task duration among cyclists (mean = 26.40 s) was significantly longer than that among non-cyclists (mean = 9.03 s) (U = 74.0, $n_{cyclist}$ = 19 $n_{non-cyclist}$ = 26, p < .001); medial-lateral short term scaling exponents were significantly smaller among cyclists (mean = 1.802) than those among non-cyclists (mean = 1.882) (U = 96.0, $n_{cyclist}$ = 18, $n_{non-cyclist}$ = 23, p = .004); medial-lateral long-term scaling exponents among cyclists (mean = 0.332) were significantly smaller than those among non-cyclists (mean = 0.698)(U = 82.0, $n_{cyclist}$ = 18, $n_{non-cyclist}$ = 23, p = .001); anterior-posterior long-term scaling exponents among cyclists (mean = 0.400) were significantly smaller than those among non-cyclists (mean = 0.798)(U = 80.0, $n_{cyclist}$ = 18, $n_{non-cyclist}$ = 23, p < .001); medial-lateral crossover points among cyclists (mean = 2.397) were significantly larger than those among non-cyclists (mean = 2.203)(U = 315.0, $n_{cyclist}$ = 18, $n_{non-cyclist}$ = 23, p = .005); and anterior-posterior crossover points among cyclists (mean = 2.377) were significantly larger than those among non-cyclists (mean = 2.126)(U = 341.0, $n_{cyclist}$ = 18, $n_{non-cyclist}$ = 23, p < .001).

Table 3 presents Pearson correlations between type of physical activity and balance measures. Cycling for racing or rough terrain was significantly associated with left leg stance medial-lateral sway range (r = .363, p < .05) and anterior-posterior short term scaling exponent

**Table 2. Statistical results of balance assessment comparing cyclists and non-cyclists.**

| | Total | | Cyclist | | Non-Cyclist | | | |
|---|---|---|---|---|---|---|---|---|
| | M or N | SD or % | M or N | SD or % | M or N | SD or % | U Statistic | *p*-value |
| **Eyes closed** | | | | | | | | |
| Sway Area ($A$) | 449.73 | 602.25 | 277.50 | 142.54 | 575.58 | 767.67 | 300.00 | .223 |
| Average Sway Velocity ($v$) | 19.68 | 7.35 | 8.03 | 5.52 | 20.89 | 8.35 | 280.00 | .448 |
| Stability Parameter ($S$) | 25.32 | 9.82 | 22.91 | 7.47 | 27.09 | 11.04 | 292.00 | .301 |
| Medial-Lateral Sway Range ($r_X$) | 17.52 | 12.78 | 13.83 | 4.81 | 20.21 | 15.91 | 303.00 | .198 |
| Anterior-Posterior Sway Range ($r_Y$) | 33.35 | 12.70 | 31.13 | 9.99 | 34.97 | 14.34 | 266.00 | .662 |
| Medial-Lateral Short Term Scaling Exponent ($\alpha_{ST,X}$) | 1.29 | 0.16 | 1.28 | 0.13 | 1.29 | 0.18 | 268.0 | .629 |
| Medial-Lateral Long Term Scaling Exponent ($\alpha_{LT,X}$) | 0.37 | 0.13 | 0.31 | 0.09 | 0.41 | 0.14 | 124.0 | .005 |
| Anterior-Posterior Long Term Scaling Exponent ($\alpha_{LT,Y}$) | 0.41 | 0.11 | 0.43 | 0.09 | 0.40 | 0.12 | 287.0 | 0.358 |
| Medial-Lateral Crossover Point ($\tau_{Cross,X}$) | 2.74 | 0.20 | 2.79 | 0.18 | 2.69 | 0.21 | 331.0 | .054 |
| Anterior-Posterior Crossover Point ($\tau_{Cross,Y}$) | 2.65 | 0.19 | 2.63 | 0.19 | 2.66 | 0.20 | 213.0 | .435 |
| **Eyes open** | | | | | | | | |
| Sway Area ($A$) | 262.06 | 314.60 | 253.23 | 374.34 | 268.52 | 270.55 | 310.00 | .148 |
| Average Sway Velocity ($v$) | 13.10 | 2.90 | 12.05 | 2.92 | 13.86 | 2.68 | 360.00 | .009 |
| Stability Parameter ($S$) | 16.81 | 3.87 | 15.67 | 3.98 | 17.63 | 3.64 | 342.00 | .029 |
| Medial-Lateral Sway Range ($r_X$) | 12.64 | 8.10 | 10.48 | 6.22 | 14.21 | 9.01 | 328.00 | .063 |
| Anterior-Posterior Sway Range ($r_Y$) | 25.87 | 7.41 | 26.44 | 8.04 | 25.45 | 7.05 | 239.00 | .854 |
| Medial-Lateral Short Term Scaling Exponent ($\alpha_{ST,X}$) | 1.22 | 0.13 | 1.23 | 0.12 | 1.23 | 0.14 | 273.0 | .550 |
| Medial-Lateral Long Term Scaling Exponent ($\alpha_{LT,X}$) | 0.38 | 0.16 | 0.35 | 0.14 | 0.40 | 0.17 | 205.0 | .334 |
| Anterior-Posterior Long Term Scaling Exponent ($\alpha_{LT,Y}$) | 0.48 | 0.15 | 0.53 | 0.18 | 0.45 | 0.11 | 330.0 | .056 |
| Medial-Lateral Crossover Point ($\tau_{Cross,X}$) | 2.73 | 0.25 | 2.74 | 0.24 | 2.71 | 0.27 | 279.0 | .462 |
| Anterior-Posterior Crossover Point ($\tau_{Cross,Y}$) | 2.67 | 0.25 | 2.65 | 0.27 | 2.69 | 0.24 | 225.0 | .613 |
| **Right leg stance** | | | | | | | | |
| Sway Area (A) | 2992.27 | 3912.37 | 3358.62 | 3575.65 | 2705.57 | 4213.66 | 144.00 | .098 |
| Average Sway Velocity ($v$) | 103.59 | 70.83 | 107.99 | 73.65 | 100.14 | 70.01 | 189.00 | .636 |
| Stability Parameter ($S$) | 152.67 | 129.08 | 148.74 | 117.20 | 155.75 | 140.21 | 193.00 | .713 |
| Medial-Lateral Sway Range ($r_X$) | 44.23 | 31.39 | 49.37 | 21.40 | 40.21 | 37.41 | 113.00 | .014 |
| Anterior-Posterior Sway Range ($r_Y$) | 74.52 | 49.04 | 83.98 | 49.46 | 67.12 | 48.51 | 124.00 | .029 |
| Task Duration ($T$) | 14.65 | 19.03 | 23.68 | 22.20 | 7.58 | 12.61 | 97.00 | .004 |
| Medial-Lateral Short Term Scaling Exponent ($\alpha_{ST,X}$) | 1.88 | 0.10 | 1.90 | 0.06 | 1.86 | 0.12 | 246.0 | .306 |
| Medial-Lateral Long Term Scaling Exponent ($\alpha_{LT,X}$) | 0.57 | 0.42 | 0.38 | 0.32 | 0.72 | 0.43 | 82.0 | .001 |
| Anterior-Posterior Long Term Scaling Exponent ($\alpha_{LT,Y}$) | 0.61 | 0.43 | 0.40 | 0.29 | 0.78 | 0.46 | 103.0 | .006 |
| Medial-Lateral Crossover Point ($\tau_{Cross,X}$) | 2.31 | 0.22 | 2.37 | 0.19 | 2.26 | 0.24 | 287.0 | .036 |
| Anterior-Posterior Crossover Point ($\tau_{Cross,Y}$) | 2.32 | 0.37 | 2.37 | 0.19 | 2.28 | 0.47 | 307.0 | .009 |
| **Left leg stance** | | | | | | | | |
| Sway Area ($A$) | 1956.96 | 1763.19 | 2067.78 | 1318.19 | 1870.23 | 2071.67 | 143.00 | .093 |
| Average Sway Velocity ($v$) | 88.92 | 49.35 | 80.61 | 19.93 | 95.42 | 63.40 | 191.00 | .674 |
| Stability Parameter ($S$) | 114.47 | 72.29 | 106.11 | 33.23 | 121.02 | 92.45 | 168.00 | .306 |
| Medial-Lateral Sway Range ($r_X$) | 34.98 | 16.12 | 43.55 | 15.80 | 28.27 | 13.13 | 93.00 | .003 |
| Anterior-Posterior Sway Range ($r_Y$) | 67.14 | 32.61 | 78.02 | 30.82 | 58.63 | 32.06 | 121.00 | .024 |
| Task Duration ($T$) | 16.65 | 20.67 | 26.40 | 21.56 | 9.03 | 16.69 | 74.00 | < .001 |
| Medial-Lateral Short Term Scaling Exponent ($\alpha_{ST,X}$) | 1.88 | 0.08 | 1.91 | 0.06 | 1.86 | 0.09 | 265.0 | .128 |
| Medial-Lateral Long Term Scaling Exponent ($\alpha_{LT,X}$) | 0.54 | 0.38 | 0.33 | 0.22 | 0.70 | 0.41 | 82.0 | .001 |
| Anterior-Posterior Long Term Scaling Exponent ($\alpha_{LT,Y}$) | 0.62 | 0.38 | 0.40 | 0.28 | 0.80 | 0.37 | 80.0 | < .001 |
| Medial-Lateral Crossover Point ($\tau_{Cross,X}$) | 2.29 | 0.23 | 2.40 | 0.16 | 2.20 | 0,25 | 315.0 | .005 |
| Anterior-Posterior Crossover Point ($\tau_{Cross,Y}$) | 2.24 | 0.23 | 2.38 | 0.17 | 2.13 | 0.22 | 341.0 | < .001 |

Note: M = mean, N = sample size, SD = standard deviation

**Table 3. Correlates of balance by physical activity type.**

| | | Cycle Leisure | Race Cycle | Swim Comp | Swim Leisure | Walk | HI Aerobic | Other Aerobic | Weights | Conditioning | Floor Ex | Dance | Jog | Golf | Row |
|---|---|---|---|---|---|---|---|---|---|---|---|---|---|---|---|
| Eyes Open | Velocity | -.304* | -0.110 | -0.107 | 0.108 | -0.118 | -0.141 | -0.188 | 0.048 | -.474** | -0.050 | -0.059 | -0.080 | 0.110 | -0.171 |
| | Stability Parameter | -0.272 | -0.102 | -0.104 | 0.154 | -0.102 | -0.164 | -0.158 | -0.004 | -.423** | 0.018 | -0.045 | -0.073 | 0.044 | -0.188 |
| Eyes Closed | Medial-Lateral Long Term Scaling Exponent | -.373* | -0.269 | .370* | 0.060 | 0.257 | -0.197 | 0.202 | 0.064 | -0.107 | 0.093 | -0.050 | -0.040 | 0.096 | -0.200 |
| | Medial-Lateral Crossover Point | 0.178 | 0.177 | 0.152 | 0.144 | 0.196 | 0.175 | -0.167 | -0.208 | -0.044 | 0.212 | 0.062 | 0.235 | 0.083 | 0.087 |
| Left Leg Single Stance | Medial-Lateral Sway Range | .410** | .363* | -0.131 | -0.222 | 0.017 | -0.036 | -0.253 | 0.007 | 0.114 | 0.006 | 0.191 | -0.050 | -0.206 | .355* |
| | Anterior-Posterior Sway Range | 0.227 | 0.229 | -0.081 | -0.230 | -0.001 | -0.016 | -0.166 | -0.015 | 0.036 | -0.078 | 0.098 | 0.054 | -0.015 | 0.294 |
| | Task Duration | .344* | 0.104 | -0.115 | -0.124 | 0.063 | 0.107 | -0.220 | -0.008 | 0.209 | -0.023 | 0.239 | -0.022 | -0.157 | 0.076 |
| | Anterior-Posterior Short Term Scaling Exponent | -.357* | -.326* | 0.050 | 0.144 | 0.054 | 0.010 | 0.093 | -0.088 | -0.113 | 0.156 | -0.119 | 0.016 | 0.213 | -.485** |
| | Medial-Lateral Long Term Scaling Exponent | -.413** | -0.204 | 0.023 | .477** | -.420** | -0.164 | .335* | 0.107 | -0.207 | -0.232 | -0.133 | -0.173 | -0.089 | -0.163 |
| | Anterior-Posterior Long Term Scaling Exponent | -.475** | -0.237 | 0.261 | .320* | -0.307 | 0.008 | .337* | 0.027 | -0.240 | -0.160 | -0.154 | -0.153 | -0.035 | -0.212 |
| | Medial-Lateral Crossover Point | .339* | 0.168 | -0.036 | -.407** | .310* | 0.195 | -.461** | -0.121 | 0.191 | 0.140 | 0.119 | 0.129 | -0.019 | 0.094 |
| | Anterior-Posterior Crossover Point | .460** | 0.210 | -0.190 | -0.284 | 0.230 | -0.028 | -0.298 | -0.067 | 0.243 | 0.090 | 0.216 | 0.105 | 0.021 | 0.164 |
| Right Leg Single Stance | Medial-Lateral Sway Range | 0.133 | 0.229 | -0.081 | 0.111 | 0.016 | 0.085 | 0.181 | 0.190 | 0.161 | -0.105 | 0.098 | -0.114 | -0.175 | 0.290 |
| | Anterior-Posterior Sway Range | 0.144 | .327* | 0.085 | 0.172 | -0.023 | 0.190 | 0.122 | 0.146 | 0.179 | 0.042 | 0.043 | -0.069 | -0.151 | 0.266 |
| | Task Duration | .379* | 0.073 | -0.099 | -0.041 | -0.092 | 0.163 | -0.114 | 0.010 | .436** | -0.040 | -0.089 | -0.047 | -0.122 | 0.110 |
| | Anterior-Posterior Short Term Scaling Exponent | -0.111 | -0.089 | 0.121 | 0.023 | -0.046 | 0.007 | 0.098 | -0.115 | -0.097 | 0.223 | -0.164 | -0.028 | 0.157 | -0.228 |
| | Medial-Lateral Long Term Scaling Exponent | -.373* | -0.192 | 0.105 | -0.031 | -0.011 | -0.133 | -0.086 | 0.116 | -0.225 | 0.003 | -0.091 | -0.106 | 0.083 | -0.165 |
| | Anterior-Posterior Long Term Scaling Exponent | -.434** | -0.241 | 0.129 | 0.035 | 0.017 | -0.093 | -0.053 | 0.014 | -0.296 | 0.096 | -0.021 | 0.069 | 0.150 | -0.272 |
| | Medial-Lateral Crossover Point | 0.195 | 0.077 | -0.174 | -0.086 | 0.292 | 0.188 | 0.129 | 0.064 | .476** | 0.214 | 0.053 | 0.123 | -0.139 | 0.059 |
| | Anterior-Posterior Crossover Point | 0.111 | 0.054 | -0.063 | -0.117 | 0.279 | 0.037 | -0.052 | 0.228 | .465** | 0.243 | -0.017 | -0.050 | -0.175 | 0.070 |

*. Correlation is significant at the 0.05 level (2-tailed).

**. Correlation is significant at the 0.01 level (2-tailed).

(r = -.326, p < .05); and right leg single leg stance anterior-posterior sway range (r = .327, p < .05). Recreational cycling was significantly correlated with eyes open sway velocity (r = -.304, p < .05); eyes closed medial-lateral long term scaling exponent (r = -.373, p < .05); left leg single stance medial-lateral sway range (r = .410, p < .01), task duration (r = .344, p < .05), anterior-posterior short term scaling exponent (r = -.357, p < .05), medial-lateral long term scaling exponent (r = -.413, p < .01), anterior-posterior long term scaling exponent (r = -.475, p < .01), medial-lateral crossover point (r = .339, p < .05), anterior-posterior crossover point (r = .460, p < .01); and right leg single stance task duration (r = .379, p < .05),

medial-lateral long term scaling exponent (r = -.373, p < .05), anterior-posterior long term scaling exponent (r = -.434, p < .01). Competitive swimming is significantly correlated with eyes closed medial-lateral long term scaling exponent (r = .370, p < .05) and recreational swimming is significantly correlated with left leg single stance medial-lateral long term scaling exponent (r = .477, p < .01), anterior-posterior long term scaling exponent (r = .320, p < .05), medial-lateral crossover point (r = -.407, p < .01). Walking is significantly correlated with medial-lateral long term scaling exponent (r = -.420, p < .01), medial-lateral crossover point (r = .310, p < .05). Aerobic activity (other) was significantly correlated with medial-lateral long term scaling exponent (r = .355, p < .05), anterior-posterior long term scaling exponent (r = .337, p < .05), medial-lateral crossover point (r = -.461, p < .01). Conditioning on a machine such as a stationary bike is significantly correlated with eyes open velocity (r = -.474, p < .01) and stability parameter (r = -.423, p < .01); and right leg single leg task duration (r = .436, p < .01), medial-lateral crossover point (r = .476, p < .01), and anterior-posterior crossover point (r = .465, p < .01). Lastly, rowing is significantly associated with left leg single stance medial-lateral sway range (r = .355, p < .05) and anterior-posterior short term scaling exponent (r = -.485, p < .01). No other variables were significantly related to balance measures.

## Conclusion

The results of this study suggest that older adults who bicycle perform significantly better on measures of balance, specifically on measures of double-leg stance with eyes open and single-leg stance tasks, than older adults who do not cycle. Additionally, we show that older cyclists engage in more physical activity than their non-cycling counterparts.

While our overarching finding of improved balance aligns with previous research [15, 16], the directionality of improved balance (i.e., medial-lateral sway versus anterior-posterior sway) differs from previous research [14]. Sway velocity among cyclists was smaller than that among non-cyclists in eyes open conditions. This suggests cyclists use a more tightly regulated postural control strategy that may relate to higher stability [26]. Given the role of vision in postural control [27–29] it is likely that removal of visual input increased center of pressure variability in eyes closed trials to the extent that significant differences could no longer be detected between cycling and non-cycling groups. DFA analysis detected few differences between cyclists and non-cyclists during double-stance tasks. However, we did observe a cycling-related decrease in medial-lateral long-term scaling exponents during the eyes closed condition. While both were smaller than 0.5 and thus indicative of anti-persistent behavior over longer time scales, the smaller exponent among cyclists suggests the presence of long-term correlations that are more negative in comparison to those of non-cyclists. This in turn suggests that, among cyclists, COP movements in one direction are more likely to be followed by COP movements in the opposite direction compared to non-cyclists, a finding that could be attributable to greater medial-lateral stability among cyclists. This is consistent with similar studies that have reported age-related increases in scaling exponents, suggesting that young adults use a more complex and more tightly regulated postural control strategy [24, 30].

Task duration among cyclists in single leg stance (right and left) was significantly longer than non-cyclists indicating increased postural stability. Cyclists also exhibited larger medial-lateral and anterior-posterior ranges of center of pressure movement in comparison to non-cyclists for single-leg stance trials. Although larger center of pressure movement is generally associated with less stable and more loosely regulated postural control [31, 32], it may be that cyclists were more comfortable with (or capable of) correcting large deviations of the body's center of mass [26]. In contrast, given their shorter trial durations, non-cyclists likely ended

single-leg tasks before the center of mass was allowed to deviate as far as that observed in the cycling group. DFA analysis revealed cycling-related decreases in long-term scaling exponents for both medial-lateral and anterior-posterior sway directions, as well as cycling-related increases in medial-lateral and anterior-posterior crossover points. Furthermore, mean long-term scaling exponents among cyclists were smaller than 0.5, while those among non-cyclists were larger than 0.5. This latter result indicates that non-cyclists do not transition into an anti-persistent region of COP control as is observed among cyclists, again suggesting a cycling-related increase in stability. The smaller crossover points observed among non-cyclists indicates a tendency to transition into the long-term region at earlier time scales. While this transition is not accompanied by a shift from persistent to anti-persistent control, it may suggest a strategy among non-cyclists to improve stability by transitioning earlier to a postural control scheme with reduced positive correlations. These results are again consistent with the notion of age-related decreases in COP complexity and stability [24, 30]. While our study included only older adults and did not specifically investigate age effects, the differences we observed between cyclists and non-cyclists suggests that cycling could reduce or postpone the effects of normal aging on postural control.

Single-leg stance is particularly important in falls prevention; evidence suggests that 40% of gait cycle is comprised of the single-leg stance [33]. Additionally, single-leg stance has been able to predict falls [34, 35] and is a valid measure of discerning older adults who fall and who do not fall among community-dwelling older adults [36]. Rissel and colleagues (2013) found that single-leg stance balance improved for older adult participants in a cycling intervention [13]. These findings coupled with our findings suggest cycling may uniquely contribute to improving one's balance.

Balance differences between the cycling and the non-cycling group may in part be explained by physical activity differences between the two groups. Physical activity was significantly higher among cyclists compared to non-cyclists. Evidence suggests physical activity may mitigate the risk for falls [4]. However, evidence suggests that balance-specific activities such as tai chi are important to increase balance [11, 37] and that walking (steps) alone do not significantly contribute to increased balance [37]. For example, in a study comparing a tai chi intervention to a brisk walking intervention, it was found that participants in the tai chi intervention had greater improvements in single leg balance than the brisk walking group [37]. Moreover, our study found that self-reported cycling frequency was significantly correlated to various balance measures more so than other types of physical activity. This evidence coupled with our findings suggests that cycling may be contributing to the increased balance rather than cyclists' increased levels of steps.

The study findings are limited by potential confounding factors such as physical activity levels, age, gender and other demographic differences. These demographic characteristics of the cyclists align with previous research [38], and therefore may need intentional efforts to overcome. We were unable to control for the potentially confounding factors due to the small and non-normally distributed samples. However, this study is strengthened by the use of objective physical activity and validated balance measures. Due to an observational cross sectional study design, selection bias is likely present as those who have poor balance may not choose to cycle or frequent the same areas as cyclists. Recruiting an overall active sample of adults and randomly assigning to a cycling intervention would eliminate selection bias for future studies.

Cycling may be a low-impact, simple way to improve balance and reduce falls among older adults. Results from this investigation suggest older adults who cycle demonstrate better postural control in two-leg stance trails and can adjust to maintain balance single-leg stance trials longer than older adults who do not cycle. Findings from the present study supports how exercise relates to balance and suggests that cycling may uniquely contribute to balance. Future

research in this area can assess the impact of frequency, duration, and type of cycling activities on balance and falls risk through the implementation of a randomized cycling intervention.

## Supporting information

**S1 File. Survey, Garmin, balance data.**
(XLSX)

## Author Contributions

**Conceptualization:** Joseph S. Lightner, Amanda Grimes.

**Data curation:** Maya Baughn, Victor Arellano, Joseph S. Lightner, Amanda Grimes, Gregory King.

**Formal analysis:** Joseph S. Lightner, Amanda Grimes, Gregory King.

**Funding acquisition:** Maya Baughn, Victor Arellano.

**Investigation:** Maya Baughn, Victor Arellano, Brieanna Hawthorne-Crosby, Amanda Grimes, Gregory King.

**Methodology:** Maya Baughn, Victor Arellano, Brieanna Hawthorne-Crosby, Joseph S. Lightner, Amanda Grimes, Gregory King.

**Project administration:** Maya Baughn, Joseph S. Lightner, Amanda Grimes.

**Resources:** Maya Baughn, Victor Arellano, Amanda Grimes, Gregory King.

**Software:** Brieanna Hawthorne-Crosby, Joseph S. Lightner, Gregory King.

**Supervision:** Maya Baughn, Joseph S. Lightner, Amanda Grimes, Gregory King.

**Validation:** Joseph S. Lightner, Amanda Grimes, Gregory King.

**Visualization:** Maya Baughn, Joseph S. Lightner, Amanda Grimes, Gregory King.

**Writing – original draft:** Maya Baughn, Victor Arellano, Brieanna Hawthorne-Crosby, Amanda Grimes, Gregory King.

**Writing – review & editing:** Maya Baughn, Victor Arellano, Joseph S. Lightner, Amanda Grimes, Gregory King.

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
