## [Decision Letter · Decision Letter 0]

25 Aug 2022

PONE-D-22-22818Physical activity, balance, and bicycling in older adultsPLOS ONE

Dear Dr. Arellano,

Thank you for submitting your manuscript to PLOS ONE. After careful consideration, we feel that it has merit but does not fully meet PLOS ONE’s publication criteria as it currently stands. Therefore, we invite you to submit a revised version of the manuscript that addresses the points raised during the review process.

As you will see, both Reviewers like the topic and motivation of your work, but also see significant problems. I invite you to submit a revision that addresses those problems.

We look forward to receiving your revised manuscript.

Kind regards,

Thomas A Stoffregen, PhD

Academic Editor

PLOS ONE

Journal Requirements:

Additional Editor Comments:

Both reviewers see merit in the motivation for this study. Both are open to the idea that cycling might improve balance and, thereby, reduce fall risk. However, each Reviewer identifies serious flaws with the submitted manuscript. I hope that a major revision can remediate those flaws. If you do submit a revision (which I encourage), then in your Response to Reviewers please be certain explicitly to respond to each of the Reviewers' comments, and the issues raised. Also, make clear where Readers can find relevant changes in the manuscript.

Reviewers' comments:

Reviewer's Responses to Questions

**Comments to the Author**

1. Is the manuscript technically sound, and do the data support the conclusions?

Reviewer #1: Partly

Reviewer #2: No

2. Has the statistical analysis been performed appropriately and rigorously? 

Reviewer #1: Yes

Reviewer #2: No

3. Have the authors made all data underlying the findings in their manuscript fully available?

Reviewer #1: Yes

Reviewer #2: Yes

4. Is the manuscript presented in an intelligible fashion and written in standard English?

Reviewer #1: Yes

Reviewer #2: Yes

5. Review Comments to the Author

Reviewer #1: Ameliorating fall risk is important, and various forms of exercise are associated with reduction in fall risk. Too, a great many researchers assume that reductions in the spatial dynamics of postural sway will reduce fall risk. The authors compared recreational cyclists vs. non-cyclists, and found some differences in sway. While the motivation is strong, critical details of the study are problematic.

I am seeing two problems with the study.

1. There is a high likelihood of self-selection bias. We have no evidence of causality. It seems more than likely that older adults with poor balance would choose to not ride bicycles. Thus, it is likely that the groups in the current study differed before and independent of actual bicycle usage. At minimum, this issue must be addressed, explicitly, in a revision. What really is needed is a true experiment—that is, a study in which individuals are randomly assigned to cycling and non-cycling groups. This might be done, for example, by buying bicycles and giving them to (randomly selected) half of a sample of non-cyclists. Pre-post measures of balance, combined with data on actual bike usage (obtainable from most cell phones, nowadays) would allow the authors to make statements about cause and effect, which they cannot do now.

2. The authors have analyzed only the spatial dynamics of balance. Decades of basic research have made clear that spatial dynamics (e.g., sway velocity) differ qualitatively from the temporal dynamics of movement. Abundant evidence demonstrates that a wide variety of clinical conditions are associated with changes in the temporal dynanmics of movement, often independent of changes in spatial dynamics (e.g., Lin et al., 2008; Stergiou & Decker, 2011; Yu et al., 2013). The existing data can be subjected to a new analysis of some measure of the temporal dynamics of sway (e.g., detrended fluctuation analysis, recurrence quantification analysis, sample entropy) using validated software routines that are readily available (e.g., in MatLab). I recommend new analyses be conducted, and the results added to the revision. Doing this can only enhance the contribution of the study.

Lin D, Seol H, Nussbaum MA, Madigan ML. Reliability of COP-based postural sway measures and age-related differences. Gait and Posture 2008;28: 337–42.

Stergiou, N., & Decker, L. M. (2011). Human movement variability, nonlinear dynamics, and pathology: Is there a connection? Human Movement Science, 30, 869-888..

Yu, Y., Chung, H.-C., Hemingway, L., & Stoffregen, T. A. (2013). Standing body sway in women with and without morning sickness in pregnancy. Gait & Posture, 37, 103-107.

Reviewer #2: PLOS ONE review

The association between cycling and falls among older adults is an interesting, and possibly a counter-intuitive one. Further research is definitely needed. A focus on older participants builds on the available literature.

Sample – people that bicycled at least once per week during non-winter months are not particularly “cyclists”. It is very low exposure – were these people more active prior to participation in the study? What were their baseline and previous physical activity levels?

There are some significant differences between the cycling and non-cycling group, particularly with their physical activity levels. Given that the cycling appears to be a very small part of the overall activity, it is impossible to say that the cycling has led to the differences. It could be any of the other activities.

A number of between group differences were found for some of the balance variables. Why some and not others if there’s a genuine effect of cycling? The authors propose that “a more tightly regulated postural control strategy that may relate to higher stability” but this finding differs from previous research- why? Again, this may well be due to generally higher physical activity levels among the cyclists.

While I am favourable towards the hypothesis, the analysis does not separate the cycling from other physical activity to allow the authors to suggest that “…cycling may be contributing to the increased balance rather than cyclists’ increased levels of steps”

Abstract

Word missing “cyclists are significantly more physically ACTIVE than…”

6. PLOS authors have the option to publish the peer review history of their article (what does this mean?). If published, this will include your full peer review and any attached files.

Reviewer #1: No

Reviewer #2: No

---

## [Author Response · Author response to Decision Letter 0]

11 Oct 2022

Please see "Response to Reviewers" file.

---

## [Decision Letter · Decision Letter 1]

17 Oct 2022

Physical activity, balance, and bicycling in older adults

PONE-D-22-22818R1

Dear Dr. Arellano,

We’re pleased to inform you that your manuscript has been judged scientifically suitable for publication and will be formally accepted for publication once it meets all outstanding technical requirements.

Kind regards,

Thomas A Stoffregen, PhD

Academic Editor

PLOS ONE

Additional Editor Comments (optional):

Reviewers' comments:

Reviewer's Responses to Questions

**Comments to the Author**

1. If the authors have adequately addressed your comments raised in a previous round of review and you feel that this manuscript is now acceptable for publication, you may indicate that here to bypass the “Comments to the Author” section, enter your conflict of interest statement in the “Confidential to Editor” section, and submit your "Accept" recommendation.

Reviewer #1: All comments have been addressed

Reviewer #2: All comments have been addressed

2. Is the manuscript technically sound, and do the data support the conclusions?

Reviewer #1: Yes

Reviewer #2: Yes

3. Has the statistical analysis been performed appropriately and rigorously? 

Reviewer #1: Yes

Reviewer #2: Yes

4. Have the authors made all data underlying the findings in their manuscript fully available?

Reviewer #1: Yes

Reviewer #2: Yes

5. Is the manuscript presented in an intelligible fashion and written in standard English?

Reviewer #1: Yes

Reviewer #2: Yes

6. Review Comments to the Author

Reviewer #1: (No Response)

Reviewer #2: The authors have responded to the reviewer's comments. The additional data and analyses have improved the manuscript.

7. PLOS authors have the option to publish the peer review history of their article (what does this mean?). If published, this will include your full peer review and any attached files.

Reviewer #1: No

Reviewer #2: No

---

## [Editor Report · Acceptance letter]

1 Dec 2022

PONE-D-22-22818R1 

Physical activity, balance, and bicycling in older adults 

Dear Dr. Arellano:

I'm pleased to inform you that your manuscript has been deemed suitable for publication in PLOS ONE. Congratulations! Your manuscript is now with our production department. 

Kind regards, 

on behalf of

Dr. Thomas A Stoffregen 

Academic Editor

PLOS ONE